# Cancer diagnostic service use in people living with HIV in South Africa: A cross-sectional study

Victor Olago[1,2]*, Gideon Nimako[2,3], Lina Bartels[4], Julia Bohlius[4,5,6], Tafadzwa Dhokotera[4,5,6,7], Matthias Egger[4,8], Elvira Singh[1,2], Mazvita Sengayi-Muchengeti[1,2,9]

1 National Health Laboratory Service (NHLS), National Cancer Registry (NCR), Johannesburg, South Africa, 2 Faculty of Health Sciences, Division of Epidemiology and Biostatistics, School of Public Health, University of the Witwatersrand, Johannesburg, South Africa, 3 Industrialization, Science, Technology and Innovation Hub, The NEPAD Planning and Coordinating Agency, Johannesburg, South Africa, 4 Institute of Social and Preventive Medicine (ISPM), University of Bern, Bern, Switzerland, 5 Swiss Tropical and Public Health Institute, Allschwil, Switzerland, 6 University of Basel, Basel, Switzerland, 7 Graduate School for Cellular and Biomedical Sciences, University of Bern, Bern, Switzerland, 8 Centre for Infectious Disease Epidemiology and Research (CIDER), School of Public Health and Family Medicine, University of Cape Town, Cape Town, South Africa, 9 South African DSI-NRF Centre of Excellence in Epidemiological Modelling and Analysis (SACEMA), Stellenbosch University, Stellenbosch, South Africa

* VictorO@nicd.ac.za

**Data Availability Statement:** We do not have the permission to share the data due to NHLS Research Material and Data Access Policy. To access the data a request must be made through

## Abstract

### Objective

The objective of this study was to map place of cancer diagnosis in relation to Human Immunodeficiency Virus (HIV) care centre among people living with HIV (PLHIV) within South Africa (SA) using national laboratory database.

### Design

We linked HIV and cancer laboratory data from 2004–2014 using supervised machine-learning algorithms. We performed a cross-sectional analysis comparing province where individuals accessed their HIV care versus where they had their cancer diagnosis.

### Setting

We used laboratory test records related to HIV diagnostics and care, such as CD4 cell counts and percentages, rapid tests, qualitative Polymerase Chain Reaction (PCR), antibody and antigen tests for HIV data that was documented as HIV positive and laboratory diagnosed cancer records from SA.

### Study population

Our study population consisted of HIV records from the National Health Laboratory Service (NHLS) that linked to cancer record at the National Cancer Registry (NCR) between 2004–2014.

the NHLS Academic Affairs and Management System (https://aarms.nhls.ac.za/).

**Funding:** This project was supported by the National Institutes of Health (NIH) administrative supplement to Existing NIH Grants and Cooperative Agreements (Parent Admin Supplement) (The South African HIV/AIDS Match Study (SAM); U01AI069924 - 09, Principal Investigator (PI) Matthias Egger, co- PI Julia Bohlius) President's Emergency Plan for AIDS Relief (PEPFAR) supplement (PI Matthias Egger) and the Swiss National Science Foundation (SNSF) (The SAM, 320030_169967, PI Julia Bohlius). The National Cancer Registry (NCR) provided office space and technical support and supervision for the study. This project has received funding from the European Union's Horizon 2020 research and innovation programme under the Marie Skłodowska-Curie grant agreement No 801076, through the SSPH+ Global PhD Fellowship Programme in Public Health Sciences (GlobalP3HS) of the Swiss School of Public Health The funders had no role in study design, data collection and analysis, decision to publish, or preparation of the manuscript.

**Competing interests:** The authors have declared that no competing interests exist.

**Abbreviations:** CD4, Cluster of Differentiation 4; CDW, Corporate Data Warehouse; EC, Eastern Cape Province; ELISA, Enzyme-Linked Immunosorbent Assay; FS, Free State Province; GAU, Gauteng Province; HIV, Human Immunodeficiency Virus; IQR, Interquartile Range; KZN, Kwa-Zulu Natal Province; LIM, Limpopo Province; MPU, Mpumalanga Province; NC, Northern Cape Province; NCR, National Cancer Registry; NHLS, National Health Laboratory Service; NW, North West Province; PLHIV, People living with HIV; SA, South Africa; SAM, Study, South African HIV Cancer Match Study; WC, Western Cape Province.

### Primary and secondary outcomes

We linked HIV records from NHLS to cancer records at NCR in order to study the inherent characteristics of the population with both HIV and cancer.

### Results

The study population was 68,284 individuals with cancer and documented HIV related laboratory test. The median age at cancer diagnosis was 40 [IQR, 33–48] years for the study population with most cancers in PLHIV diagnosed in females 70.9% [n = 46,313]. Of all the PLHIV and cancer, 25% (n = 16,364 p < 0.001) sought treatment outside their province of residence with 60.7% (n = 10,235) travelling to Gauteng. KZN had 46.6% (n = 4,107) of its PLHIV getting cancer diagnosis in Gauteng. Western Cape had 95% (n = 6,200) of PLHIV getting cancer diagnosis within the province.

### Conclusions

Our results showed health systems inequalities across provinces in SA with respect to cancer diagnosis. KZN for example had nearly half of the PLHIV getting cancer diagnosis outside the province while Western Cape is able to offer cancer diagnostic services to most of the PLHIV in the province. Gauteng is getting over burdened with referral for cancer diagnosis from other provinces. More effort is required to ensure equitable access to cancer diagnostic services within the country.

## 1. Introduction

Equitable and reliable access to timely cancer diagnostic services can help reduce cancer morbidity and mortality [1–4]. Early cancer diagnosis is more likely to result in malignancies being diagnosed at a treatable stage [1,5]. In South Africa (SA), equitable and timely access to oncology services has several issues [6–8]. Access to cancer care is tiered with primary health facilities providing mostly screening and palliative care services and regional and tertiary hospitals providing cancer diagnostic and treatment services [9]. Many, but not all, provinces in SA have regional and tertiary hospitals that have the mandate to provide specialised oncology services [9]. However, oncology services have been poorly resourced with reports of failing services in some provinces [7,10]. It has been postulated that this has resulted in patients having to travel long distances and outside their home province for cancer diagnosis and care. To date, not many studies have looked at the potential geographic inequities in cancer diagnostic services among the people living with HIV (PLHIV) in SA. A study in Uganda reported on multiple barriers that patients with HIV associated malignancies have in receiving both HIV and cancer treatment [11].

Oncology services in low and middle-income countries are hindered by poor infrastructure, limited human and financial resources and lack of skilled human resources [3]. A study on breast cancer diagnosis showed that the distance from the health facility affected the stage of cancer diagnosis [12]. The further the distance one lived from the health facility the more likely one was to present at the hospital with late stage breast cancer [12]. Similarly, a study on lung cancer care in the KwaZulu-Natal (KZN) province of SA reported that geographical location was a barrier to cancer care access as most of the oncology centres were in urban areas [13].

The study also highlighted that lack of psychosocial care and resources limits access to screening, diagnosis and treatment of cancer [13]. Two other studies in KZN province, one that reviewed endoscopy services and the other that looked at waiting times for prostate cancer diagnosis, reported poor maintenance of infrastructure, lack of trained personnel and frequent disruptions, as barriers to effective and timely cancer care [14]. Although SA is an upper middle-income country in Sub Saharan Africa [15], it is not exempt from the challenges facing oncology services in the region.

In a bid to achieve the UNAIDS HIV 2020 targets of 95-95-95 [16] HIV diagnosis and care has been effectively decentralized in SA [17]. Most PLHIV are accessing HIV diagnosis, care and treatment in dispensaries, clinics and health centres in close proximity of their residence [17]. The Department of Health (DOH) in SA has a well-defined referral system in the public health infrastructure with the aim of reducing morbidity and having better health outcomes [18]. Making cancer diagnostic services more accessible may lead to better cancer treatment outcome in SA. SA, annually records over 80,000 incident cancer cases with over 50,000 cancer related deaths [19,20]. It is estimated that 19.6% of cancer cases are attributed to infections [19], while the HIV infection rate in Black cancer patients is estimated at 34% [21].

The South African HIV Cancer Match study (SAM) consists of an HIV cohort created from routinely collected public sector HIV-diagnosis and HIV-care related laboratory data. This data were linked to the cancer data from the South African National Cancer Registry (NCR) to identify HIV positive cancer cases. This created an opportunity to determine access to cancer diagnostic services for PLHIV. As such, we aimed to assess the cancer diagnostic use among the PLHIV in SA.

## 2. Methods

### 2.1 Study setting

This was a study conducted using South African public sector laboratory data from all the 9 provinces. Gauteng being the province with the capital city, is the most populous but has the smallest land surface; Northern Cape (NC) on the other hand has the largest land surface but has the smallest population [22]. The National Health Laboratory Service (NHLS) is the largest provider of pathology services in SA covering an estimated 80% of the population [23]. Its specialised divisions include the National Institute of Communicable Diseases, the National Institute of Occupational Health, the National Cancer Registry and the Corporate Data Warehouse (CDW). All the results of tests done by the network of NHLS laboratories are collated in the CDW allowing for extraction for various research purposes upon permission. The National Cancer Registry (NCR) is the primary cancer surveillance system in SA [24]. In 2011, legislation was passed that made cancer-reporting to the NCR mandatory for persons and organisations diagnosing malignancies in the country [25]. The pathology based registry is complete for cases diagnosed by histology, cytology and bone marrow aspirates and trephines but excludes cases diagnosed by clinical and other investigations. We used laboratory test records related to HIV diagnostics and care, such as CD4 cell counts and percentages, rapid tests, qualitative PCR, antibody and antigen tests for the HIV data that was documented as HIV positive and laboratory diagnosed cancer records.

### 2.2 Study design and study population

This was a cross-sectional analysis from a record linkage study [26]. Record linkage is the process of identifying records that potentially belong to the same person in different datasets or one dataset [27]. We used a support vector machine algorithm, a supervised machine learning technique to link the HIV and cancer data [28]. The linkage variables included names,

surnames, age or date of birth, episode number and folder numbers. These variables were pre-processed and standardised to allow for the linkage. Our study population consisted of HIV records from the NHLS that linked to a cancer record at the NCR, both datasets for the years 2004 to 2014. For the HIV data, we first de-duplicated the dataset to acquire individual records which we then linked to incident cancer cases from the NCR. The deduplication process involved data cleaning and standardization, names strings comparison to get the similarity weights and classification of the similarity weights using support vector machine algorithms. For patients with more than one cancer diagnosis report we used the first pathology report only, thus including incident cancer cases. For patients with HIV records in more than one province we used the province with the most records as the home province. The HIV data had prevalent HIV records. We used records where the HIV treatment date was earlier than the cancer diagnosis date. Since this was a cross-sectional study we included HIV records that had only single test performed.

## 2.3 Software

We implemented this work entirely in Python 3.6 [29], running in Anaconda (Enterprise 4) [30] using Jupyter Ipython Notebook (version 5.3.1) [31]. Python modules Pandas (version 0.25.1) [32] for data manipulation, GeoPandas (version 0.5.1) [33] for mapping geographic data, and Seaborn (version 0.9.0) [34] and Matplotlib (version 3.1.1) [35] for visualization were used.

## 2.4 Data management

It should be noted that area of home residence of cases is not available in the HIV dataset or the cancer surveillance dataset. The HIV data included information on facility, laboratory, district and province of treatment. Throughout the years, HIV testing has become more accessible resulting in all levels of health care (primary to tertiary) providing the service. In most cases, whether rural or urban, in SA, patients do not have to travel long distances to access HIV testing and care. As a result, we assumed that the province of HIV testing and care was the home province of patients. The cancer dataset also had information on the district and province of diagnosis. We used Pandas version 0.25.1 for data manipulation. We accessed the HIV and cancer datasets for cleaning, deduplication and linkage between January to December of 2018 after which we kept anonymised dataset for analysis purposes.

## 2.5 Study hypothesis, analysis and visualization

Assuming the province of HIV treatment to be the home province of the patients, we investigated whether all patients received cancer diagnosis in their home province. In order to understand the characteristics of our study population, we applied chi-square to assess the differences in the discrete groups, logistic regression to test the impact of age, gender, race and cancer type to the province of cancer diagnosis. We created tables for demographic characteristics, a tabulation of the province where the patient received HIV treatment versus where they received cancer diagnosis. We also created a table showing how age, gender and cancer type affect province of cancer diagnosis. We used GeoPandas, Seaborn and Matplotlib to create the maps marking facilities of HIV treatment and those of cancer diagnosis. We used South African choropleth maps that had demarcation of the provincial boundaries. The choropleths are available at https://www.naturalearthdata.com and are free to use [36].

## 2.6 Ethics

Ethical approval for this study was obtained from the University of the Witwatersrand Human Research Ethics Committee. Ethics certificate number M171176 was issued. Permission to conduct the study was also granted by the relevant organisations, the NCR and NHLS respectively. Seeking individual consents from patients involved in this study was not possible since this was historical data from established databases.

## 2.7 Patient and public involvement

The study is based on routinely collected laboratory data; therefore, no patients were involved in the design, conduct, reporting, or dissemination plans of our research. The results of analyses of the data will be shared with Department of Health both at the Provincial and National levels to help in allocation of resources of cancer diagnostic resources.

## 3. Results

During the 11-year period (2004–2014), 664,869 cancers were diagnosed in SA and reported to the NCR. The linkage between NCR and NHLS resulted in 115,333 records. We dropped records with HIV results marked as unknown, negative and cases that had cancer diagnosis date prior to HIV treatment. This resulted to 65,284 malignancies with a documented HIV positive result. Of the 65,284 cancer and HIV co-morbidity patients, 25% (n = 16,364 p < 0.001) sought treatment outside their province of residence. The median age at cancer diagnosis was 40 years (Interquartile range (IQR): 33–48) for the total linked population (HIV positive individuals) and 39 years (IQR: 32–47) for the proportion that sought cancer diagnosis outside their province of HIV care. Table 1 shows the demographic characteristics of our study population.

Only 43% of PLHIV from KZN sought cancer diagnosis within the province, as compared to 95% of PLHIV from Western Cape as shown in Table 2. Gauteng diagnosed the most cancer cases in the PLHIV at 42% (n = 27,660), followed by Western Cape 11.6% (n = 7,571) and Mpumalanga 9.7% (n = 6,344).

**Table 1. Demographics characteristics of the study population.**

| Characteristics | Location of cancer diagnosis | | HIV + patients with cancer diagnosis | P-value |
|---|---|---|---|---|
| | Outside of province of HIV care | Within province of HIV care | | |
| **Total Number** | 16,364(25.1%) | 48,920 (74.9%) | 65,284 | <0.001 |
| **Age [years]** | | | | |
| Median (IQR) | 39 (IQR 32,47) | 40 (IQR 33,49) | 40 (IQR 33,48) | |
| **Gender** | | | | |
| Female | 12,110 (26.2%) | 34,203 (73.9%) | 46,313 | <0.001 |
| Male | 4,254 (22.4%) | 14,708 (77.6%) | 18,962 | |
| **Ethnicity** | | | | |
| Black | 15,088 (26.2%) | 42,407 (73.8%) | 57,495 | <0.001 |
| Coloured | 484 (15.0%) | 2,732 (85.0%) | 3,216 | |
| White | 645 (21.2%) | 2,399 (78.8%) | 3,044 | |
| Asian | 69 (18.0%) | 315 (82.0%) | 384 | |
| **Blacks vs non-Blacks** | | | | |
| Black | 15,088(26.2%) | 42,407(73.8%) | 57,495 | <0.001 |
| non-Black | 1,198(18.0%) | 5,446(82.0%) | 6,644 | |

**Table 2. Province of cancer diagnosis versus province of HIV diagnosis and care, N (column percentages).** The green highlights show the proportion that accessed cancer diagnosis and HIV diagnosis and care in their home province. The blue highlights shows where GAU and WC provinces performed cancer diagnosis to PLHIV from other provinces.

| Province of HIV treatment | | EC | FS | GAU | KZN | LIM | MPU | NW | NC | WC | TOTAL |
|---|---|---|---|---|---|---|---|---|---|---|---|
| **Province of cancer diagnosis** | EC | 4,202 (80.8%) | 17 (0.3%) | 92 (0.5%) | 94 (1.1%) | 8 (0.2%) | 15 (0.2%) | 25 (0.5%) | 25 (1.5%) | 126 (1.9%) | 4,606 (7.1%) |
| | FS | 19 (0.4%) | 4,250 (82.2%) | 392 (1.9%) | 101 (1.2%) | 45 (0.9%) | 110 (1.4%) | 127 (2.8%) | 53 (3.3%) | 11 (0.2%) | 5,108 (7.8%) |
| | GAU | 260 (5%) | 579 (11.2%) | 17,682 (87.1%) | 4,104 (46.8%) | 853 (17.3%) | 2,840 (35.3%) | 1,131 (24.7%) | 85 (5.3%) | 126 (1.9%) | 27,660 (42.5%) |
| | KZN | 51 (1%) | 16 (0.3%) | 391 (1.9%) | 3,806 (43.4%) | 14 (0.3%) | 183 (2.3%) | 11 (0.2%) | 1 (0.1%) | 17 (0.3%) | 4,490 (6.9%) |
| | LIM | 3 (0.1%) | 17 (0.3%) | 322 (1.6%) | 26 (0.3%) | 3,856 (78.1%) | 174 (2.2%) | 60 (1.3%) | 3 (0.2%) | 4 (0.1%) | 4,465 (6.9%) |
| | MPU | 17 (0.3%) | 39 (0.8%) | 946 (4.7%) | 536 (6.1%) | 125 (2.5%) | 4,634 (57.6%) | 31 (0.7%) | 8 (0.5%) | 8 (0.1%) | 6,344 (9.7%) |
| | NW | 23 (0.4%) | 52 (1%) | 226 (1.1%) | 28 (0.3%) | 22 (0.4%) | 35 (0.4%) | 3,037 (66.5%) | 19 (1.2%) | 8 (0.1%) | 3,450 (5.3%) |
| | NC | 18 (0.3%) | 37 (0.7%) | 43 (0.2%) | 7 (0.1%) | 1 (0%) | 8 (0.1%) | 51 (1.1%) | 1,251 (77.5%) | 35 (0.5%) | 1,451 (2.2%) |
| | WC | 608 (11.7%) | 166 (3.2%) | 209 (1%) | 64 (0.7%) | 14 (0.3%) | 43 (0.5%) | 97 (2.1%) | 170 (10.5%) | 6,200 (94.9%) | 7,571 (11.6%) |
| | TOTAL | 5,203 (7.99%) | 5,173 (9.94%) | 20,303 (31.17%) | 8,766 (13.46%) | 4,938 (7.58%) | 8,042 (12.34%) | 1,615 (2.48%) | 4,570 (7.02%) | 6,535 (10.03%) | |

Key: EC–Eastern Cape, FS–Free States, GAU–Gauteng, KZN–Kwa-Zulu Natal, LIM–Limpopo, NW–North West, NC–Northern Cape, WC–Western Cape.

Of all the PLHIV and cancer, 25% (n = 16,864) accessed cancer diagnosis outside their province of residence with 60.7% (n = 10,235) travelling to Gauteng. KZN had 46.6% (n = 4,107) of its PLHIV getting cancer diagnosis in Gauteng. Limpopo, North West and Mpumalanga also referred a substantial number of PLHIV to Gauteng for cancer diagnosis.

With the increase in age PLHIV are less likely to travel to another province for cancer diagnosis as shown in Table 3. The table also shows that male PLHIV are less likely to travel to another province for cancer diagnosis while being Black increases their chances of traveling to another province for cancer diagnosis. PLHIV who received cancer diagnosis for Basal Cell Carcinoma, Prostate and Kidney are likely to travel to another province for cancer diagnosis. We dropped all cancers records with missing province of cancer diagnosis.

We mapped the HIV treatment centers versus the cancer diagnostic centers in SA as shown in Fig 1. Gauteng received PLHIV for cancer diagnosis from all the provinces, with most coming from KZN, Mpumalanga, Limpopo and North West as shown in Fig 2. Western Cape on the other hand received PLHIV for cancer diagnosis from Eastern Cape and Gauteng as shown in Fig 3.

## 4. Discussion

Overall, 25% of PLHIV travelled out of their home province for cancer diagnosis. There was poor access to cancer diagnosis in PLHIV in KZN such that nearly half (46.8%) of patients studied accessed cancer diagnosis in Gauteng, which is hundreds of kilometres from their HIV diagnosis/care province. In 2015, the South African Human Rights Commission submitted a petition to the National Department of Health highlighting the declining cancer care services in KZN province [7]. The petition stated that cancer patients had to wait at least five months before seeing an oncologist [7]. The wait extended to 13 months if the recommended course of

**Table 3. The association of age, gender, race and cancer type to the province of cancer diagnosis.**

| Variable | Odds ratio | P-value | 95% confidence Interval |
|---|---|---|---|
| Age | 1.014 | <0.001 | 1.013, 1.016 |
| gender (Female—base) | | | |
| Male | 1.061 | 0.019 | 1.010, 1.115 |
| Race (non-Black—base) | | | |
| Black | 0.608 | <0.001 | 0.569, 0.651 |
| Cancer | | | |
| BCC | 0.447 | <0.001 | 0.334, 0.599 |
| Bladder | 0.632 | 0.010 | 0.445, 0.896 |
| Bone | 0.667 | 0.045 | 0.449, 0.990 |
| Burkitt lymphoma | 2.009 | <0.001 | 1.498, 2.695 |
| Hodgkin lymphoma | 1.370 | 0.020 | 1.051, 1.787 |
| Kaposi Sarcoma | 1.584 | <0.001 | 1.259, 1.993 |
| Kidney | 0.548 | 0.001 | 0.386, 0.778 |
| Melanoma | 0.715 | 0.050 | 0.511, 1.000 |
| Mesothelioma | 0.512 | 0.032 | 0.278, 0.944 |
| Non Hodgkin lymphoma | 1.260 | 0.055 | 0.995, 1.596 |
| Pancreas | 0.472 | 0.003 | 0.287, 0.776 |
| Prostate | 0.450 | <0.001 | 0.340, 0.594 |
| Skin other | 0.715 | 0.035 | 0.523, 0.977 |
| Stomach | 0.711 | 0.031 | 0.522, 0.968 |
| Testis | 0.548 | 0.027 | 0.322, 0.934 |
| Thyroid | 0.682 | 0.031 | 0.481, 0.966 |
| Tongue | 1.480 | 0.053 | 0.995, 2.201 |

treatment was radiotherapy. This depicts a different picture from the guide of early cancer diagnosis that recommends treatment initiation within one month of diagnosis [5]. In KZN province, have three large oncology hospitals but they were unable to meet the cancer diagnosis demand of its residents [9]. A third (35.3%) and a quarter (24.7%) of PLHIV from Mpumalanga and North West provinces respectively travelled to Gauteng province for cancer diagnosis.

Gauteng hospitals treated more than 20% of PLHIV for cancer who had travelled from other provinces. The Gauteng province has the majority of academic hospitals, which are more likely to be well equipped and well-staffed. Other than being the province with the capital city, the province is surrounded by Free State, North West, Limpopo and Mpumalanga provinces thereby making referrals from these provinces easier as it is more accessible. However, since most provinces are referring to it, this will most likely overburden the Gauteng hospitals resulting in a problem similar to that in KZN. Already some PLHIV from Gauteng province opt to get cancer diagnostic services in Western Cape. (Table 2). SA has already in place referral guidelines for patients who require diagnostic services in upper health facilities tiers even though some patients by-pass lower hospital facilities when believe they are more likely not to receive proper health services in the facilities close to them [37].

The Western Cape Province offered cancer diagnosis to most of its PLHIV. This is a sign of better and more accessible cancer diagnostic services within the province. A survey done by the National Department of Health revealed that the spectrum and quantity of oncology services in the Western Cape province, including infrastructure and personnel, were sufficient for provincial needs [9]. The Western Cape Province is mostly urban with increased geographical

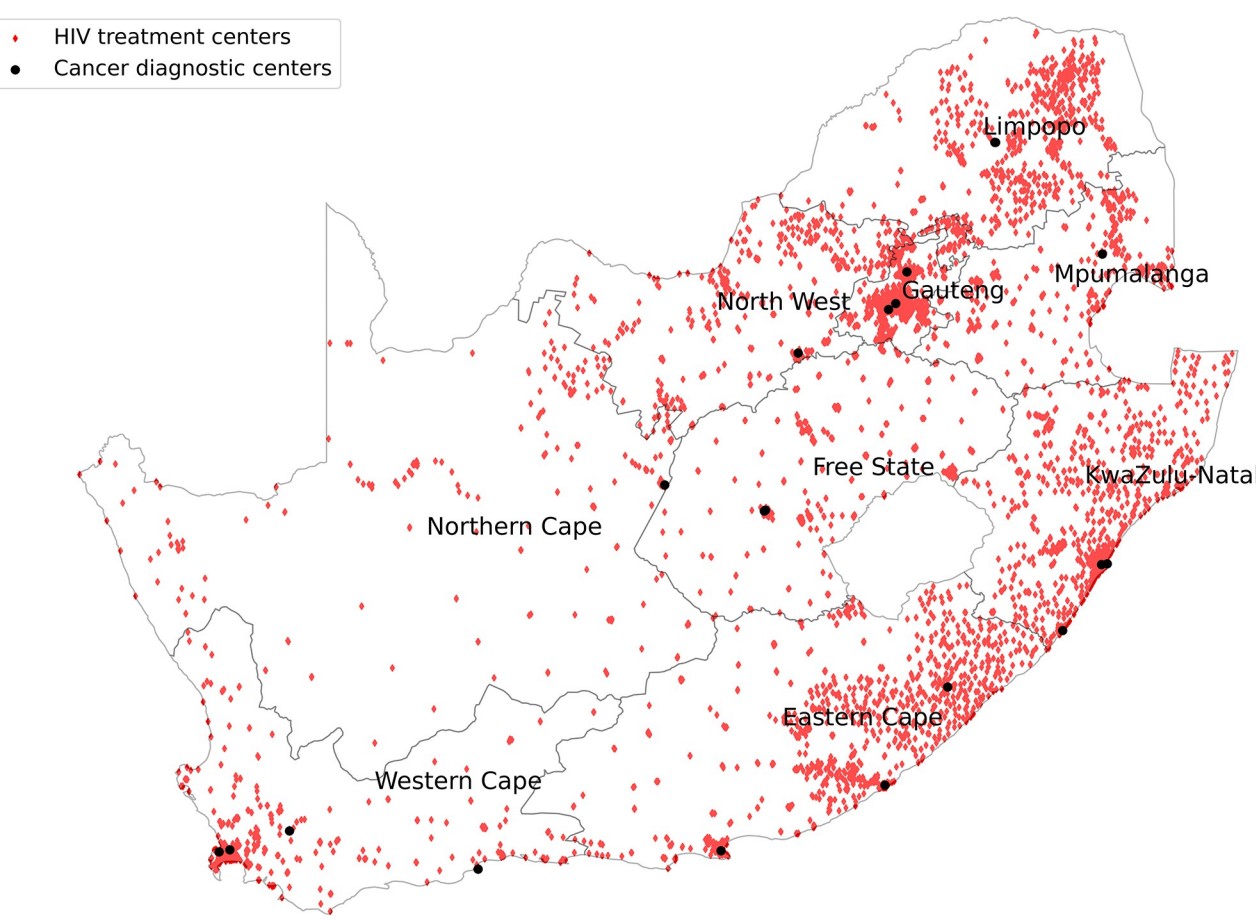

**Fig 1. HIV treatment centers versus cancer diagnostic centers.**

accessibility and efficient referral pathways [38]. However, referrals from other provinces are not common as WC is distant from other provinces compared to Gauteng province as shown in Figs 1 and 3.

Our study demonstrated major inequities with regards to access to public healthcare sector cancer diagnostic services for PLHIV in SA [39]. Assuming that the province where patients accessed HIV care was their home province, we reported that the average age at cancer diagnosis for PLHIV was 40 years with younger age groups likely to travel or be referred to another province for cancer diagnosis. We recommend further investigations to explore this effect and to determine whether the quicker diagnosis in receiving provinces resulted in down staging of cancers and a survival advantage. PLHIV who were treated for either Burkitt lymphoma, Hodgkin lymphoma, Kaposi Sarcoma, Non Hodgkin lymphoma and Tongue were more likely to get cancer diagnosis in their home province unlike the PLHIV who were treated for Basal Cell Carcinoma, Prostate, Kidney, Testis, Bladder, Bone, Melanoma, Mesothelioma, Pancreas, Thyroid and Stomach as shown in Table 3. Patients who suffered from AIDS Defining Cancers (ADC) were able to get cancer diagnosis within their home province as opposed to those who suffered from Non-AIDS Defining Cancers (NDC).

This study also demonstrated the earlier age of cancer diagnosis in PLHIV at 40 years compared 60 years which is the average age for cancer diagnosis in the South African population according the NCR data. The early cancer diagnosis age for PLHIV may be as a results of HIV related cancer policies in SA such as screening activities in the HIV positive population [40].

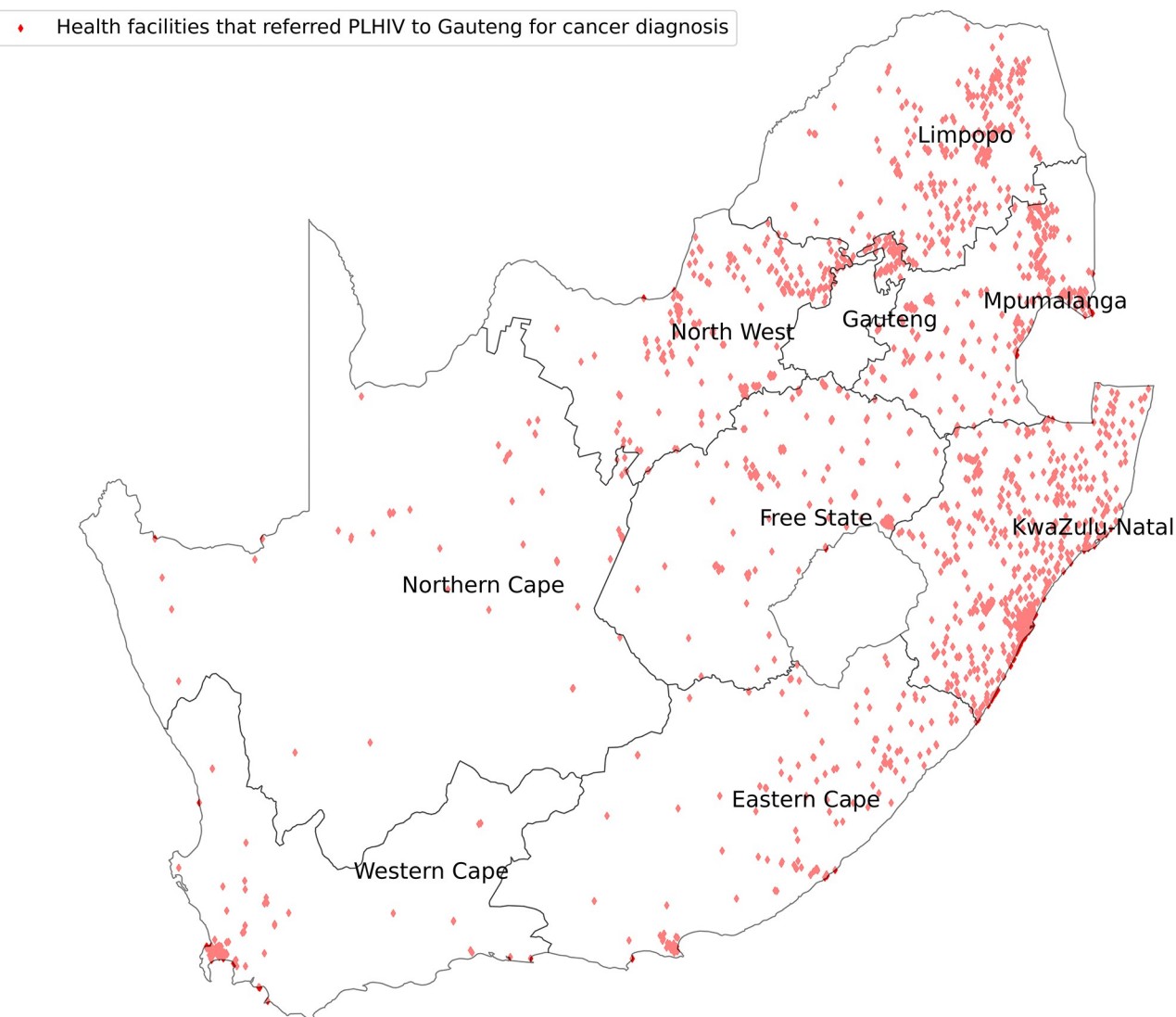

Health facilities that referred PLHIV to Gauteng for cancer diagnosis

**Fig 2. HIV treatment centers that referred their cancer patients for diagnosis to Gauteng Province.**

Black Africans made up 92% of PLHIV who travelled for cancer diagnosis and 74% were female. This reflects the gender and population group distribution of the South African HIV epidemic as reported in the South African National HIV prevalence, incidence and behaviour survey [41]. The HIV epidemic in SA affects more women than men and more Black compared to other population groups [41]. The majority of the Black population access their healthcare in the public sector compared to the White population that access care in the private sector. Black South Africans have been marginalised in the country due to socio-economic circumstances and previous political oppression [42]. This reflects a further barrier to healthcare in an already disadvantaged population who may not afford the costs associated with travelling long distances to access healthcare [43].

The 74% of patients who travelled for cancer diagnosis being female contrast to traditional SA economic migration patterns where men travelled for work opportunities [44]. Recently, Vearey *et al* reported the feminisation of migratory patterns as more women enter the workforce. However, this pattern could also reflect the pattern of health seeking behaviour in SA

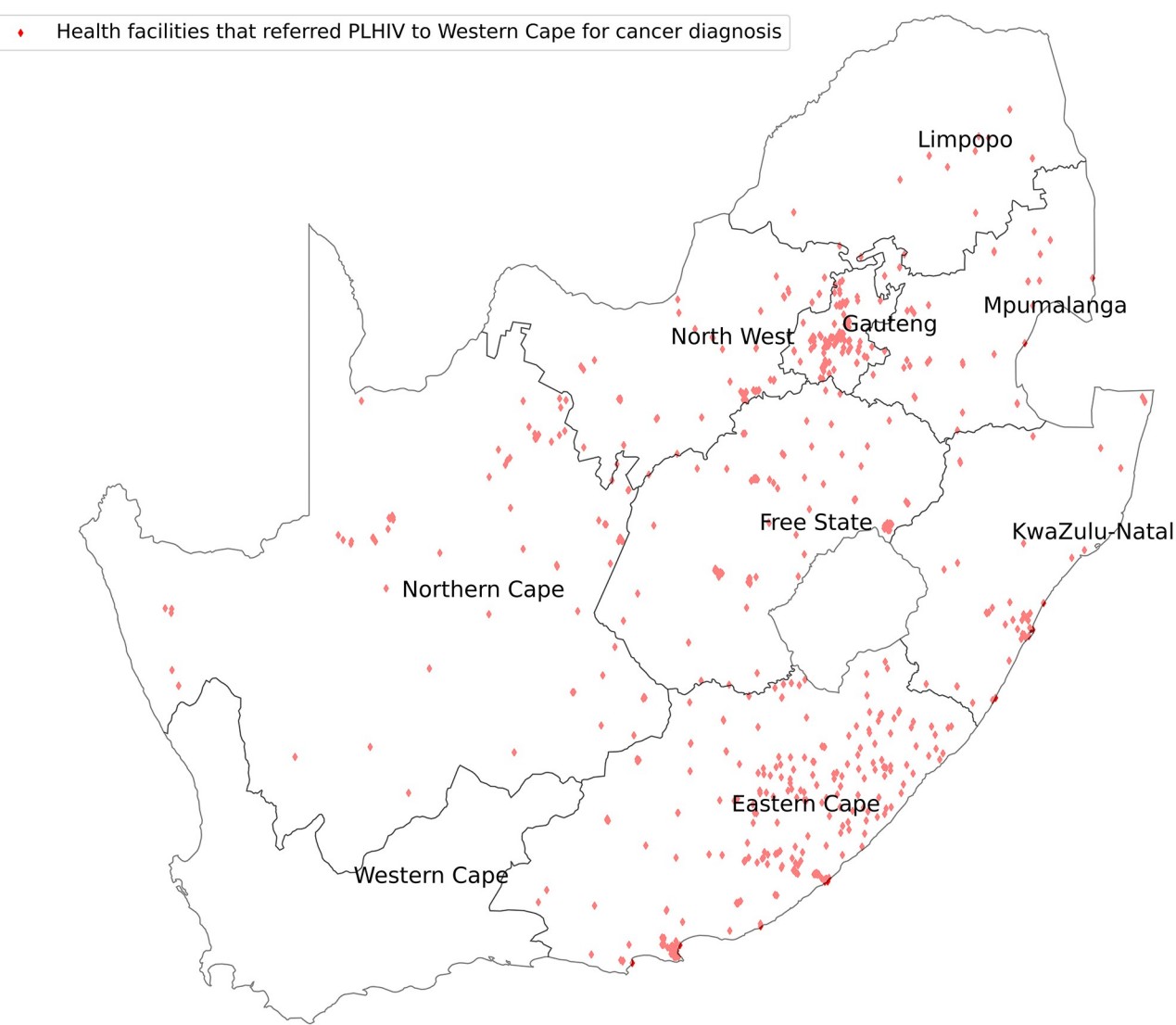

**Fig 3. HIV treatment centers that referred their cancer patients for diagnosis to Western Cape Province.**

where women are more likely to seek care for health symptoms than men [45]. Women in our study population were relatively younger compared to men. There is also a significant over representation of women (p>0.001) in our study population which is similar to HIV population in SA [41]. SA also has policies and guidelines for breast and cervical cancer screening leading to more cancers detected among women. Such screening services may have also led to better cancer diagnosis of cervix and breast cancers in their home province as the proportion that sought treatment outside their province of HIV treatment was 30% and below.

Our study was one of the few studies trying to determine accessibility of cancer diagnostic services in SA. The major limitation of our study was that it did not account for movement due to other reasons such as migration for better jobs and education. Veary *et al* asserted that internal migration in SA occurs more often due to economic opportunities than for seeking of healthcare services [45]. However, that study was conducted amongst primary healthcare users while our study assessed cancer diagnosis, which occurs at the regional and tertiary hospital level. Furthermore, diagnostic pathology reports do not provide patient's residential address.

Therefore, our HIV cohort allowed us to have a proxy for the actual residential address using the facility at which HIV was diagnosed. This also meant, we could not account for the population of PLHIV who sought HIV treatment in another province due to stigma. NCR is a cancer pathology registry, the total cancers reported may not give the density of cancer patients per cancer center, which means we could not determine the level of overcrowding at the cancer centers. Since SAM study is a virtual cohort created from data linkages, we recommend further qualitative studies which could further help understand the challenges faced by population of PLHIV and cancer.

We assert that patients in SA are forced to move province for cancer care as a result of under-resourced regional and tertiary oncology services in their home province while HIV care, accessed at primary healthcare level is abundantly available in their home province.

## 5. Conclusion

While HIV services have been decentralised and are available to patients close to where they live and work, more effort is required to ensure equitable access to oncology services within the country to optimise cancer patient outcomes. Our results showed health systems inequalities across provinces in SA with respect to cancer diagnosis. Patients who suffered from cancers that are classified as ADC's were more likely to receive their cancer diagnosis within their home province. This showed need to strengthen cancer diagnostic services particularly in under-served provinces.

## Acknowledgments

We would like to express our deep gratitude to the late Elvira Singh, the head of the South African National Cancer Registry from 2013 to 2022, for her invaluable role in making the SAM study possible.

## Author Contributions

**Conceptualization:** Gideon Nimako, Lina Bartels, Julia Bohlius, Tafadzwa Dhokotera, Matthias Egger, Elvira Singh, Mazvita Sengayi-Muchengeti.

**Methodology:** Victor Olago.

**Supervision:** Elvira Singh, Mazvita Sengayi-Muchengeti.

**Writing – original draft:** Victor Olago.

**Writing – review & editing:** Gideon Nimako, Lina Bartels, Julia Bohlius, Tafadzwa Dhokotera, Matthias Egger, Elvira Singh, Mazvita Sengayi-Muchengeti.

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
