## [Decision Letter · Decision Letter 0]

9 Oct 2023

PONE-D-23-25603Cancer diagnostic service use in people with HIV in South Africa: a cross-sectional studyPLOS ONE

Dear Dr. Olago,

Thank you for submitting your manuscript to PLOS ONE. After careful consideration, we feel that it has merit but does not fully meet PLOS ONE’s publication criteria as it currently stands. Therefore, we invite you to submit a revised version of the manuscript that addresses the points raised during the review process.

A rebuttal letter that responds to each point raised by the academic editor and reviewer(s). You should upload this letter as a separate file labeled 'Response to Reviewers'.A marked-up copy of your manuscript that highlights changes made to the original version. You should upload this as a separate file labeled 'Revised Manuscript with Track Changes'.An unmarked version of your revised paper without tracked changes. You should upload this as a separate file labeled 'Manuscript'.We look forward to receiving your revised manuscript.

Kind regards,

Ibrahim Jahun, MD, MSC, PhD

Academic Editor

PLOS ONE

Journal Requirements:

"We would like to express our deep gratitude to the late Elvira Singh, the  head of the South African National Cancer Registry from 2013 to 2022, for her invaluable role in making the SAM study possible. This project was supported by the National Institutes of Health (NIH) administrative supplement to Existing NIH Grants and Cooperative Agreements (Parent Admin Supplement) (The South African HIV/AIDS Match Study (SAM); U01AI069924 - 09, Principal Investigator (PI) Matthias Egger, co- PI Julia Bohlius) President’s Emergency Plan for AIDS Relief (PEPFAR) supplement (PI Matthias Egger) and the Swiss National Science Foundation (SNSF) (The SAM, 320030_169967, PI Julia Bohlius). The National Cancer Registry (NCR) provided office space and technical support and supervision for the study. This project has received funding from the European Union’s Horizon 2020 research and innovation programme under the Marie Skłodowska-Curie grant agreement No 801076, through the SSPH+ Global PhD Fellowship Programme in Public Health Sciences (GlobalP3HS) of the Swiss School of Public Health"

6. We note that Figures 1-3 in your submission contain map images which may be copyrighted. All PLOS content is published under the Creative Commons Attribution License (CC BY 4.0), which means that the manuscript, images, and Supporting Information files will be freely available online, and any third party is permitted to access, download, copy, distribute, and use these materials in any way, even commercially, with proper attribution. For these reasons, we cannot publish previously copyrighted maps or satellite images created using proprietary data, such as Google software (Google Maps, Street View, and Earth). For more information, see our copyright guidelines: http://journals.plos.org/plosone/s/licenses-and-copyright.

1.) You may seek permission from the original copyright holder of Figures 1-3 to publish the content specifically under the CC BY 4.0 license.  

2.) If you are unable to obtain permission from the original copyright holder to publish these figures under the CC BY 4.0 license or if the copyright holder’s requirements are incompatible with the CC BY 4.0 license, please either i) remove the figure or ii) supply a replacement figure that complies with the CC BY 4.0 license. Please check copyright information on all replacement figures and update the figure caption with source information. If applicable, please specify in the figure caption text when a figure is similar but not identical to the original image and is therefore for illustrative purposes only.

**Additional Editor Comments:**

ACADEMIC EDITOR: The manuscript's objectives are of immense value toward improving cancer management among PLHIV. However, there are areas that require clarity to ensure the contents are explicitly presented. The authors should respond to the following observations in addition to reviewers' comments highlighted below: Please complete the ethics question using the approval institutions and approval umber mentioned under “ethics” in the study methods. NA doesn’t fit well here.Please review captions of all the figures in this paper  and ensure they are align with plosone figure/table guidelines: https://journals.plos.org/plosone/s/tables and https://journals.plos.org/plosone/s/figures

Reviewers' comments:

Reviewer's Responses to Questions

**Comments to the Author**

1. Is the manuscript technically sound, and do the data support the conclusions?

Reviewer #1: Partly

Reviewer #2: Yes

2. Has the statistical analysis been performed appropriately and rigorously? 

Reviewer #1: Yes

Reviewer #2: Yes

3. Have the authors made all data underlying the findings in their manuscript fully available?

Reviewer #1: Yes

Reviewer #2: No

4. Is the manuscript presented in an intelligible fashion and written in standard English?

Reviewer #1: Yes

Reviewer #2: Yes

5. Review Comments to the Author

Reviewer #1: The study findings are important for public health actions toward improving access to cancer services. The manuscript needs to be reviewed to ensure consistency and clearly review the implications of the assumptions that form the basis for the conclusion drawn from the study. All these have been articulated in the comments below:

Study design/population:

- “For the HIV data, we first de-duplicated the dataset to acquire individual records”:

Comment: will be good to briefly outline how the deduplication was done.

- “For patients with HIV records in more than one province we used the province with the most records as the home province.”

Comment: This assumption is a bit “plausible”. The location maybe the preferred location for the client. Has the study factored the tradition where PWH travel to far locations to access care due to stigma?

Results:

- “We dropped records with HIV results marked as unknown, negative and cases that had cancer diagnosis date prior to HIV treatment”.

Comment: how about identified HIV clients who delayed treatment and had cancer prior commencing ART? During 2004 – 2014, Test and Treat wasn’t a common approach.

- “Cancer and HIV co-infected patients”

Comments: consider using “co-morbidities” since cancer is not regarded as an infection but rather a condition.

- “We dropped all cancers that did not show any association on whether a patient was diagnosed within out outside home province”.

Comment: The authors should carefully review this. It is better to report the number of such cases otherwise the decision to drop them may be perceived as bias or some manipulations to please the study objectives.

- “Table 3: The impact of age, gender, race and cancer type to the province of cancer diagnosis

Comment: consider using “association” instead of “impact”. Also no province listed in the table, please review this.

Discussions:

- The study didn’t indicate the density of cancer patients per cancer centre in the provinces. It is likely that some cancer centres are well crowded due to high rate of cancer in the province (not only among HIV clients). Overcrowding may lead to long waiting time and poor cancer care which may force some HIV clients to move to other provinces with less densely cancer centres. Please provide data if available and discuss this scenario. Otherwise consider adding this as limitation if data isn’t available.

- Also include concise recommendation based on the findings.

Limitations:

- The study didn’t identify why actually patients move to other cancer centres; therefore, the conclusions are based on assumptions. It will be good to conduct qualitative study to strengthen the findings in this study. Consider discussing this under “limitations”.

Reviewer #2: 1. The authors used commonly known terms with standard abbreviation numerous times and could have written the term in the first use abbreviated the term throughout the study e.g South Africa (SA) can be written in full in the first use and later abbreviated as SA throuought the manuscript.

2. The UNAIDS target should be updated

3. The authors used PWH (numerous times) interchangeably with PLHIV (in maps) in the article and the most commonly acceptable phrase for individuals who are diagnosed with HIV is People Living With HIV (PLHIV). The authours are advised to use PLHIV rather than PWH.

4. The authors can share the anonymised linked dataset as an appendix

6. PLOS authors have the option to publish the peer review history of their article (what does this mean?). If published, this will include your full peer review and any attached files.

Reviewer #1: No

Reviewer #2: No

---

## [Author Response · Author response to Decision Letter 0]

1 May 2024

Reviewer #1: The study findings are important for public health actions toward improving access to cancer services. The manuscript needs to be reviewed to ensure consistency and clearly review the implications of the assumptions that form the basis for the conclusion drawn from the study. All these have been articulated in the comments below:

Study design/population:

- “For the HIV data, we first de-duplicated the dataset to acquire individual records”:

Comment: will be good to briefly outline how the deduplication was done.

Author response: We added the description of the deduplication as follows “The deduplication process involved data cleaning and standardization, names strings comparison to get the similarity weights and classification of the similarity weights using support vector machine algorithms.” Page 7 1st paragraph

- “For patients with HIV records in more than one province we used the province with the most records as the home province.”

Comment: This assumption is a bit “plausible”. The location maybe the preferred location for the client. Has the study factored the tradition where PWH travel to far locations to access care due to stigma?

Author response: By using facility address as a proxy to the PLHIV address we could not account for the population of PLHIV who sought HIV treatment in another province due to stigma.

Results:

- “We dropped records with HIV results marked as unknown, negative and cases that had cancer diagnosis date prior to HIV treatment”.

Comment: how about identified HIV clients who delayed treatment and had cancer prior commencing ART? During 2004 – 2014, Test and Treat wasn’t a common approach. 

Author response: The scope of the study was to assess the cancer diagnostic service use among PLHIV, the inclusion criteria required HIV diagnosis comes prior to cancer diagnosis.

- “Cancer and HIV co-infected patients”

Comments: consider using “co-morbidities” since cancer is not regarded as an infection but rather a condition.

Author response: We changed the word to co-morbidity. Page 9 1st paragraph

- “We dropped all cancers that did not show any association on whether a patient was diagnosed within out outside home province”.

Comment: The authors should carefully review this. It is better to report the number of such cases otherwise the decision to drop them may be perceived as bias or some manipulations to please the study objectives.

Author response: We updated the wording to “We dropped all cancers records with missing province of cancer diagnosis.”. Page 11 2nd Paragraph

- “Table 3: The impact of age, gender, race and cancer type to the province of cancer diagnosis

Comment: consider using “association” instead of “impact”. Also no province listed in the table, please review this. 

Author response: Table 3 shows the results for a binary logistic regression indicating factors associated with patients receiving cancer diagnosis outside of their home provinces. Province is the outcome of the regression. We have changed the word impact to association. Page 11

Discussions:

- The study didn’t indicate the density of cancer patients per cancer centre in the provinces. It is likely that some cancer centres are well crowded due to high rate of cancer in the province (not only among HIV clients). Overcrowding may lead to long waiting time and poor cancer care which may force some HIV clients to move to other provinces with less densely cancer centres. Please provide data if available and discuss this scenario. Otherwise consider adding this as limitation if data isn’t available.

Author response: We added a limitation sentence as follows “NCR is a cancer pathology registry, the total cancers reported may not give the density of cancer patients per cancer center, which means we could not determine the level of overcrowding at the cancer centers.”. Page 15 2nd Paragraph

Limitations:

- The study didn’t identify why actually patients move to other cancer centres; therefore, the conclusions are based on assumptions. It will be good to conduct qualitative study to strengthen the findings in this study. Consider discussing this under “limitations”.

Author response: We added this as a limitation “Since SAM study is a virtual cohort created from data linkages, we recommend further qualitative studies which could further help understand the challenges faced by population of PLHIV and cancer.” Page 15 2nd Paragraph

Reviewer #2: 1. The authors used commonly known terms with standard abbreviation numerous times and could have written the term in the first use abbreviated the term throughout the study e.g South Africa (SA) can be written in full in the first use and later abbreviated as SA throuought the manuscript.

Author response: We changed South Africa to SA after the first instance.

2. The UNAIDS target should be updated

Author response: We updated the UNAIDS targets. Page 5 2nd Paragraph

3. The authors used PWH (numerous times) interchangeably with PLHIV (in maps) in the article and the most commonly acceptable phrase for individuals who are diagnosed with HIV is People Living With HIV (PLHIV). The authours are advised to use PLHIV rather than PWH.

Author response: We updated the wording to PLHIV throughout the manuscript.

4. The authors can share the anonymised linked dataset as an appendix

Author response: We do not have permission to publicly share the data.

---

## [Decision Letter · Decision Letter 1]

20 May 2024

Cancer diagnostic service use in people living with HIV in South Africa: a cross-sectional study

PONE-D-23-25603R1

Dear Dr. Olago,

We’re pleased to inform you that your manuscript has been judged scientifically suitable for publication and will be formally accepted for publication once it meets all outstanding technical requirements.

Kind regards,

Ibrahim Jahun, MD, MSC, PhD

Academic Editor

PLOS ONE

Additional Editor Comments (optional):

Reviewers' comments:

Reviewer's Responses to Questions

**Comments to the Author**

1. If the authors have adequately addressed your comments raised in a previous round of review and you feel that this manuscript is now acceptable for publication, you may indicate that here to bypass the “Comments to the Author” section, enter your conflict of interest statement in the “Confidential to Editor” section, and submit your "Accept" recommendation.

Reviewer #1: All comments have been addressed

Reviewer #2: All comments have been addressed

2. Is the manuscript technically sound, and do the data support the conclusions?

Reviewer #1: Yes

Reviewer #2: Yes

3. Has the statistical analysis been performed appropriately and rigorously? 

Reviewer #1: Yes

Reviewer #2: Yes

4. Have the authors made all data underlying the findings in their manuscript fully available?

Reviewer #1: No

Reviewer #2: Yes

5. Is the manuscript presented in an intelligible fashion and written in standard English?

Reviewer #1: Yes

Reviewer #2: Yes

6. Review Comments to the Author

Reviewer #1: All comments raised have fully been addressed and the paper is fit to proceed for publication. However, not all data are made available and this is well understood. Some of the data are confidential and require special request.

Reviewer #2: (No Response)

7. PLOS authors have the option to publish the peer review history of their article (what does this mean?). If published, this will include your full peer review and any attached files.

Reviewer #1: **Yes: **Jahun Ibrahim

Reviewer #2: **Yes: **Lactatia Motsuku

---

## [Editor Report · Acceptance letter]

23 May 2024

PONE-D-23-25603R1 

PLOS ONE

Dear Dr. Olago, 

I'm pleased to inform you that your manuscript has been deemed suitable for publication in PLOS ONE. Congratulations! Your manuscript is now being handed over to our production team.

Kind regards, 

on behalf of

Dr. Ibrahim Jahun 

Academic Editor

PLOS ONE